# Study of SPRC Impact Resistance Based on the Weibull Distribution and the Response Surface Method

**DOI:** 10.3390/polym14112281

**Published:** 2022-06-03

**Authors:** Song Chen, Ziling Xu, Zeli Liu, Chen Wang, Jiuhong Jiang

**Affiliations:** College of Civil Engineering, Architecture and Environment, Hubei University of Technology, Wuhan 430068, China; 102010881@hbut.edu.cn (S.C.); 101910580@hbut.edu.cn (Z.X.); 101910585@hbut.edu.cn (Z.L.); 102010862@hbut.edu.cn (C.W.)

**Keywords:** silica fume, polyvinyl alcohol fiber, impact resistance, Weibull distribution model, response surface method

## Abstract

Silica-fume–polyvinyl-alcohol-fiber-reinforced concrete (SPRC) is a green and environmentally friendly composite material incorporating silica fume and polyvinyl alcohol fiber into concrete. To study the impact resistance of SPRC, compressive-strength and drop hammer impact tests were conducted on SPRC with different silica-fume and polyvinyl-alcohol-fiber contents. The mechanical and impact resistance properties of the SPRC were comprehensively analyzed in terms of the compressive strength, ductility ratio and impact-energy-dissipation variation. Based on the impact resistance of the SPRC, the impact life of SPRC with different failure probabilities was predicted by incorporating the Weibull distribution model, and an impact damage evolution equation for SPRC was established. The impact life of SPRC under the action of silica-fume content, polyvinyl-alcohol-fiber content and failure probability was analyzed in depth by the response surface method (RSM). The research results show that, when the content of silica fume is 10% and the content of polyvinyl alcohol fiber is 1%, the compressive strength and impact resistance of SPRC are the best. The RSM response model can effectively predict and describe the impact life of SPRC specimens under the action of three factors.

## 1. Introduction

Concrete is widely used in engineering as an integral part of the construction field [1,2]. For example, concrete is widely used in bridge structures, pile foundations, aircraft runways, dams and offshore platforms. However, these structures are subjected to various dynamic loads during their service, such as traveling loads [3,4,5], aircraft landings [6], waves [7], earthquakes [8], and other dynamic loading effects. These unavoidable human [9,10,11] and natural factors [12] present serious challenges to the safety of concrete structures. Therefore, it is important to study the impact resistance of concrete structures under various impact loads for disaster prevention and mitigation.

Currently, the problem of how to enhance the impact resistance of concrete has been widely studied by researchers. To enhance impact resistance, some researchers have incorporated fibers in concrete [13]. Examples include steel fibers [14,15], carbon fibers [16], recycled glass fibers [17], polypropylene fibers [18], and natural plant fibers [19]. Some researchers have found that adding different aggregates can enhance the impact resistance of concrete. For example, Wu et al. [16] found that larger and higher strength coarse aggregates can improve the impact resistance of concrete. Zhang et al. [20] found that bauxite coarse aggregates are beneficial for improving the mechanical properties and impact resistance of cement composites. Mustafa et al. [21] concluded that the impact strength of concrete increases with the percentage of plastic waste replacing sand. Vatin [22] found that the use of asphalt-mixed aggregates can improve the impact power of concrete.

Silica fume is the industrial waste produced by ferroalloying plants. Incorporation of silica fume in concrete can reduce the amount of cement and save mineral resources. The reuse of silica fume can improve the environment, save energy, reduce emissions and reduce secondary pollution. Importantly, silica fume has high activity, pozzolanic activity and a microaggregate filling reaction, which can improve the mechanical properties and durability of concrete [23]. Silica fume is commonly used in UHPC to improve mechanical properties, and the cement replacement ratio of silica fume is typically within 20% [24]. Although Luan et al. [25] found that the drying shrinkage of UHPC increased with the increase of silica fume content, some scholars have found that the incorporation of a small amount of silica fume has many advantages [26,27,28,29,30,31]. Poon et al. [26] found that, with the incorporation of silica fume, the compressive strength and chloride ion penetration resistance of concrete were improved. Hassani et al. [27] found that, with the incorporation of silica fume, the mechanical properties of pavement-panel concrete improved significantly. Alaloul et al. [28] found that the incorporation of silica fume compensates for the loss of mechanical properties due to the incorporation of rubber. Liu et al. [29] compared the test results of concrete with silica fume incorporated, with concrete without silica fume, and found that the strength of the concrete increased with the incorporation of the silica fume. Zhao et al. [30] incorporated silica fume into concrete and showed through a series of experiments that the impact resistance of concrete was improved with the incorporation of silica fume. Shafaghat et al. [31] used electron microscopy to analyze the internal structure of silica-fume concrete. The analysis showed that silica fume improved the microstructure of concrete during the hydration of cement with a simultaneous pozzolanic reaction.

Polyvinyl alcohol (PVA) fiber is a synthetic fiber with acid resistance, alkali resistance, high strength and high modulus. As a reinforcing fiber, it has the advantages of high deformation capacity, superior tensile properties and excellent fracture properties, which have been recognized by the material-science and engineering community [32,33,34]. Li et al. [35] found that the concrete fracture-damage process becomes longer with the incorporation of polyvinyl alcohol fibers. Zhang et al. [36] found that the incorporation of polyvinyl alcohol fibers enhanced the splitting tensile strength and flexural strength of concrete. Qiu et al. [37] found that polyvinyl alcohol fibers effectively retarded the fatigue damage of concrete. Kim et al. [38] placed polyvinyl alcohol fiber concrete in a harsh environment for a period of time and scanned its permeable pores and carbonation depth using electron microscopy, and the results showed that polyvinyl alcohol fiber concrete exhibited a better durability performance. Afroughsabet et al. [39] found that, with the incorporation of polyvinyl alcohol fibers, the dry shrinkage of concrete during precuring significantly decreased. Chen et al. [40] found that the incorporation of polyvinyl alcohol fibers significantly enhanced the blast resistance of concrete. Zhao et al. [41] placed polyvinyl-alcohol-fiber concrete in a freeze–thaw environment and subjected samples to a sulfate attack in a seawater environment. They found that the concrete durability was significantly enhanced with the incorporation of polyvinyl alcohol fibers.

Some scholars have improved the mechanical properties and durability of concrete by mixing fiber and mineral materials [42,43,44,45]. Mosavinejad et al. [42] prepared ultrahigh-performance concrete with a high content of silica fume and short-cut polyvinyl alcohol fibers and concluded that, for compressive strength, the optimum ratio of silica fume to cement is 0.3, a further increase in silica fume content leads to a decrease in compressive strength, and the incorporation of PVA fibers reduces the diffusion and penetration of chloride ions. Prakash et al. [43] studied the mechanical properties and resistance to chloride ion penetration of UHPC containing silica fume and finely ground slag. Satnam et al. [44] found that mixing scrap tire rubber and silica fume can improve the impact resistance of concrete. Suruchi Mishra et al. [45] studied the impact resistance of ultrahigh-performance fiber-reinforced concrete mixed with silica fume and fiber components, and they found that the UHPFRC impact response depends heavily on the energy and velocity of the impact mass, specimen size, support stiffness, test type, and even the definition of damage. However, there are few in-depth analyses of the impact-damage process and impact life of concrete mixed with mineral materials and fibers. The purpose of this study is to predict the impact life of SPRC with different failure probabilities with the Weibull distribution model. The SPRC impact-damage evolution equation was established to study the damage evolution trend of SPRC impact life under repeated impact. The effects of silica-fume content, polyvinyl-alcohol-fiber content and failure probability on SPRC impact life were analyzed with the response surface method (RSM).

## 2. Materials and Methods

### 2.1. Materials

The cement was P.O 42.5 grade ordinary silicate cement from Wuhan Xinhua Company. The aggregates were from Wuhan. The coarse aggregate was ordinary graded gravel with particle size of 5 to 25 mm. The fine aggregate was Grade II medium sand with continuous grading and a fineness modulus of 2.83. The fly ash was Grade I fly ash from Henan Hengyuan New Materials Co. Ltd. The silica fume was produced by Henan Yixiang New Materials Co. Ltd. Polyvinyl alcohol fiber was produced by Kuraray Japan. The detailed parameters of silica fume, fly ash, cement and polyvinyl alcohol fiber are shown in Table 1. Polyvinyl alcohol fiber and silica fume are shown in Figure 1. The water-reducing agent was a polycarboxylic-acid high-performance water-reducing agent. The test water was ordinary tap water.

### 2.2. Concrete Mixing Ratio Design

The SPRC was based on C40 concrete (the cube compressive strength of 150mm side length was 40 MPa) as the base mix ratio. The 100 mm × 100 mm × 100 mm cubed specimens were taken for the compressive-strength test, and three specimens were made from each group. Cylindrical Φ 150 mm × 63 mm specimen test blocks were taken for the drop-hammer impact tests, and six specimens were made from each group. The water–binder ratio was 0.43, the content of the silica fume (0%, 5%, 10%, 15%) was varied by cement weight, and the content of polyvinyl alcohol fibers (0%, 0.5%, 1%, 1.5%) was varied by the volume of the entire mixture. The water-reducing agent content was 1 kg/m^3^. The mixing proportion is shown in Table 2.

### 2.3. Specimen Production and Test Methods

The specimens were prepared by an HJS-60 type twin-shaft concrete test mixer. Considering that polyvinyl alcohol fibers tend to agglomerate, the fibers were torn and dispersed by hand before being added to the mixer. In the mixing process, the first step was to add gravel and sand and mix for 1 min; then, cement, fly ash and silica fume were added and mixed for 1 min; finally, part of the water and the water-reducing agent were added and mixed for 2 min. The second step was to add the treated polyvinyl alcohol fiber evenly into the mixer and to continue to mix for 2 min after adding; then, the remaining water and water-reducing agent were added and mixing was continued for 2 min. To avoid uneven distribution of polyvinyl alcohol fiber in the concrete, the mixing time was extended appropriately. After mixing, the mix was poured into molds for full pounding. After pounding, the specimens were placed in a cool place for 24 h to demold, and the specimens were stored in a standard maintenance room for maintenance treatment.

The compressive-strength tests were performed according to the standards specified in GB/T 50081-2019 [46] and CECS 13:2009 [47]. The compressive-strength test of the cube specimens with a size of 100 mm × 100 mm × 100 mm was carried out by using the DYE-2000S microcomputer servo pressure testing machine. The specific operation method of the compression test was as follows: after the end of specimen maintenance, the specimen was promptly removed, excess water on the surface of the specimen was wiped off, and the appearance of the specimen was verified to meet the requirements. As the concrete strength was C40, according to the specification, the loading speed of this test was 0.5 MPa/s. After the specimen was destroyed, the test was stopped and the test results were recorded.

The impact test method adopted was the drop-hammer impact test method recommended by the American Concrete Association ACI 544 standard [48]. The method has the advantages of simple operation and relatively low test-condition requirements. This test adopts a CECS13-2009 drop-hammer impact testing machine to test the impact resistance of the specimen, as shown in Figure 2. The test method was as follows: a Φ 150 mm × 63 mm cylindrical specimen was placed in four baffles on the chassis, the specimen was 5 mm from the baffle, and a Φ 63 mm steel ball was placed on the specimen. The height of the hammer drop of the test device was fixed at the standard height, and the distance between the center of mass of the hammer and the surface of the steel ball on the test block was 500 mm. The electromagnetic switch that controls the drop hammer has a counter connected to it. The button on the electromagnetic switch controls the drop hammer for impact, and the counter records the number of impacts with each press of the button. As the number of impacts increased, when the first visible crack appeared in the specimen was regarded as the first crack, and the number of impacts at this time was recorded. When the specimen was impacted and fully cracked such that the specimen touched the three surrounding baffles, this was regarded as the final crack, and the number of impacts was recorded at this time.

## 3. Results

### 3.1. Compressive Strength of SPRC

The compressive-strength test results and compressive-strength changes are shown in Figure 3. The error bars in the figure are the standard deviations, and the error bars indicate the dispersion of the data; smaller error bars indicate smaller data dispersion. When the silica fume content was 10%, the overall growth rate of the compressive strength of SPRC was the highest. This is because of the small silica-fume size and its microaggregate filling effect, which can play a role in filling the small pores inside the cementitious composites. The incorporation of silica fume reduces the porosity and improves the internal structure. The interfacial transition zone of cement-matrix composites is often the weak region of their internal structure due to the large number of Ca(OH)_2_ crystals attached to the interfacial transition zone, which leads to the increased porosity and looser network structure in this region. Silica fume has high volcanic ash activity. The SiO_2_ in silica fume can react with Ca(OH)_2_ crystals produced by cement hydration to generate a C-H-S gel by a secondary hydration reaction, which reduces the pore volume and pore distribution in the interfacial transition zone and improves the network structure in this region, increasing the compressive strength.

With the increase in polyvinyl alcohol fiber content, the compressive-strength growth rate of SPRC was also significantly improved. This is because, when the specimen is subjected to load, the stress is often formed at the crack tip inside the specimen. With continuous loading, the stress at this tip also increases, leading to the extension of cracks inside the specimen, thus forming more cracks and eventually causing damage to the specimen. The polyvinyl alcohol fiber bridging effect exists, and fiber incorporation will form a complex force system in the concrete. This causes the concrete to be subjected to an external load, and polyvinyl alcohol fiber will form a reverse stress at the crack tip to offset part of the energy generated by the external load, thereby delaying the growth and extension of cracks so that the tolerance has been increased, and the strength has been improved.

However, a high silica-fume content and a high polyvinyl-alcohol-fiber content are not always better. When the content of silica fume exceeds 10%, the compressive-strength growth rate of the SPRC begins to decline. This is because the specific surface area of the silica-fume particles is large. When the silica-fume content is too large, silica-fume particles will adsorb more free water in cement composites. This reduces the content of water involved in the cement hydration reaction, and the hydration degree of the cement-based composites decreases, resulting in a downward trend in the compressive strength with a continuous increase in silica-fume content. When the polyvinyl-alcohol-fiber content is more than 1%, the compressive-strength growth rate of the SPRC also shows a downward trend, which is due to excessive fiber content that makes the cementitious material unable to be fully wrapped, which causes stress defects inside the specimen, resulting in compressive strength decreases with increasing polyvinyl alcohol fiber content.

### 3.2. SPRC Impact Resistance

The strength of the impact resistance of SPRC is reflected by the impact energy consumption W and ductility ratio λ. The equation for impact energy consumption W is shown in Equation (1), and the equation for the ductility ratio λ is shown in Equation (2). The test results for the number of impacts that caused the first visible cracks $N_{1}$ and the number of impacts that caused the ultimately failure of the specimens $N_{2}$ are shown in Table 3. The SPRC impact damage pattern is shown in Figure 4.
(1)$$W=N_{2}mgh$$

In Equation (1): W—impact energy consumption (J);

$N_{2}$—the number of impacts that caused the ultimate failure of specimens (times);

m—impact hammer quality (kg), 4.5 kg;

g—gravitational acceleration (m/s^2^), 9.81 m/s^2^;

h—drop height (m), 0.5 m.
(2)$$\lambda =\frac{N_{2}-N_{1}}{N_{1}}$$

In Equation (2): λ—Ductility ratio, $N_{1}$—the number of impacts to cause the first visible cracks, and $N_{2}$—the number of impacts that caused the ultimate failure of the specimens.

As seen from Table 3, due to the large discreteness of concrete itself, the discreteness of the results of the number of impacts to cause the first visible cracks $N_{1}$ and the number of impacts that caused the ultimate failure of the specimens $N_{2}$ is also large, and the results of six specimens in each group were averaged. According to Equations (1) and (2), the impact energy consumption W and ductility ratio λ were calculated. The results are shown in Table 4, and the change in impact energy consumption W is shown in Figure 5. The error bars in the figure are the standard deviations, and the error bars indicate data dispersion: smaller error bars indicate smaller data dispersion.

Table 4 shows that the ductility ratio of SPRC is significantly improved. Compared with the benchmark concrete, SPRC often experiences multiple drop-hammer impacts before the ultimate failure, which indicates that the combination of silica fume and polyvinyl alcohol fiber can enhance the ductility of SPRC. Figure 5 shows that, with the increase in silica-fume content, the impact energy consumption of SPRC first increases and then decreases, and when the silica fume content is at 10%, the impact energy consumption reaches the maximum.

After the specimen is subjected to repeated drop-hammer impacts, the internal cracks will extend and expand after the impact load. With the increasing number of impacts, the damage deterioration inside the concrete will gradually intensify, and the specimen will eventually experience failure damage. Figure 5 shows that, with the increase in polyvinyl alcohol fiber content, the impact energy consumption of SPRC first increases and then decreases, and when the polyvinyl-alcohol-fiber content is at 1%, the impact energy consumption reaches the maximum. When the silica-fume content was 10% and the polyvinyl alcohol fiber content was 1%, the impact resistance of SPRC was the best and was 154.32% higher than that of the benchmark concrete.

### 3.3. Impact Resistance Analysis of SPRC Based on the Weibull Distribution Model

#### 3.3.1. SPRC Impact Life Analysis

Due to a large number of irregular microcracks, micropores and other defects in concrete, the damage to a concrete structure under repeated drop-hammer impacts has great randomness. Impact damage can be regarded as a kind of random distribution variable that must conform to a certain statistical law. The Weibull distribution model [49,50,51] has been widely used for life prediction in the failure assessment of brittle materials. Therefore, with the number of final cracks of the SPRC as the original data, the Weibull distribution model was selected to explore the impact life changes of the SPRC under different failure probabilities. The Weibull distribution model is a three-parameter function proposed by Waloddi Weibull in the study of material fatigue life, as shown in Equation (3):(3)$$f\left(x\right)=\frac{\beta }{\eta }{\left(\frac{x-x_{0}}{\eta }\right)}^{\beta -1}\exp\left[-{\left(\frac{x-x_{0}}{\eta }\right)}^{\beta }\right]\;\;\;\left(x\geq x_{0}\right)$$

In Equation (3): β—Shape factor;

η—Scale factor;

$x_{0}$—Location parameters $\left(x_{0}\geq 0\right)$.

Due to the similarity of the failure mechanism between the impact failure and fatigue failure of specimens, the impact life N also obeys the Weibull distribution, but the three parameters of the original function need to be redefined. Let the original parameters β=b, $\eta =N_{a}-N_{0}$ and $x_{0}=N_{0}$; then, the specimen impact life N can be expressed by the following probability density function, Equation (4):(4)$$f\left(N\right)=\frac{b}{N_{a}-N_{0}}{\left(\frac{N-N_{0}}{N_{a}-N_{0}}\right)}^{b-1}\exp\left[-{\left(\frac{N-N_{0}}{N_{a}-N_{0}}\right)}^{b}\right]\;\;\;\left(\infty >N\geq N_{0}\right)$$

In Equation (4): b—Weibull shape parameters;

$N_{0}$—Minimum life parameter;

$N_{a}$—Characteristic life parameters.

Its cumulative distribution function $F\left(N_{p}\right)$ is expressed by Equation (5):(5)$$F\left(N_{p}\right)=P_{1}\left(N<N_{p}\right)=\int_{N_{0}}^{N_{p}}f\left(N\right)dN=1-\exp\left[-{\left(\frac{N_{p}-N_{0}}{N_{a}-N_{0}}\right)}^{b}\right]$$

In Equation (5), $P_{1}\left(N<N_{p}\right)$ indicates the probability of being less than a certain value $N_{p}$. The value of the function corresponding to the cumulative distribution function $F\left(N_{p}\right)$ is called the cumulative probability of failure, from which the reliability function $P_{2}$ can be expressed as:(6)$$P_{2}=P_{1}\left(N>N_{p}\right)=1-F\left(N_{p}\right)=\exp\left[-{\left(\frac{N_{p}-N_{0}}{N_{a}-N_{0}}\right)}^{b}\right]$$

After determining the survival rate $P_{2}$, the impact life $N_{p}$ under the corresponding failure probability $P_{1}$ can be calculated from Equation (6). Considering the safety and reliability of the specimen in the service process and the simplified calculation, the minimum life parameter $N_{0}$ is set to 0, and the two-parameter Weibull probability density function is obtained. The function expression is expressed as Equation (7):(7)$$f\left(N\right)=\frac{b}{N_{a}}{\left(\frac{N}{N_{a}}\right)}^{b-1}\exp\left[-{\left(\frac{N}{N_{a}}\right)}^{b}\right]\;\;\;\left(\infty >N\geq N_{0}\right)$$

Then, the failure probability function $P_{1}$ for $N>N_{p}$ is shown in Equation (8):(8)$$P_{1}=1-\exp\left[-{\left(\frac{N}{N_{a}}\right)}^{b}\right]$$

The survival rate function $P_{2}$ is calculated by Equation (9):(9)$$P_{2}=1-P_{1}=\exp\left[-{\left(\frac{N}{N_{a}}\right)}^{b}\right]$$

Taking the natural logarithm twice for both sides of Equation (9), Equation (10) is obtained.
(10)$$\ln\left[\ln\left(\frac{1}{P_{2}}\right)\right]=b\ln\left(\frac{1}{N_{a}}\right)+b\ln\left(N\right)$$

Let $y=\ln\left[\ln\left(\frac{1}{P_{2}}\right)\right],x=\ln\left(N\right),a=b\ln N_{a}$ and $N_{a}=\exp\left[-(a/b)\right]$; then, we can obtain Equation (11):(11)Y=a+bX

Equation (10) can be used to test whether the test data obey the distribution law of the Weibull probability density function, and the test data are subjected to linear regression analysis to find the parameters a and b. After linear regression analysis of the test data, whether there is a good linear relationship between Y and X is tested by combining the two parameters. If there is, the Weibull probability density function can reasonably predict and describe the impact life of SPRC under different failure probabilities. If not, the Weibull probability density function is not applicable to the impact life analysis of SPRC.

To verify whether the impact life of specimens conforms to the Weibull distribution, there should be a good performance relationship between $\ln\left[\ln\left(1/P_{2}\right)\right]$ and $\ln\left(N\right)$. The survival rate $P_{2}$ needs to be calculated first, and the calculation method is shown in Equation (12):(12)$$P_{2}=1-\frac{i}{m+1}$$

In Equation (12): m—total number of specimens per group and i—experimental data arranged in order from small to large.

Through Equation (9) to Equation (12), with X as the abscissa and Y as the ordinate, the corresponding regression parameters a and b are obtained by linear fitting of the data, as shown in Table 5.

From Table 5, the maximum value of the regression coefficient $R^{2}$ for the concrete with compound silica fume and polyvinyl alcohol fiber is 0.97532. Most of the regression coefficients $R^{2}$ of the group are greater than 0.9, and a small number are less than 0.9, but greater than 0.82022. The linear regression fitting results are good, and the test results conform to the distribution law of the Weibull probability density function, which indicates that Y=a+bX is valid.

Equation (13) can be obtained by deformation according to Equation (8) to Equation (12). The impact life of the SPRC under different failure probabilities $P_{1}$ is analyzed and estimated by Equation (13):(13)$$N=\exp\left\{\frac{\ln\left[\ln\left(1/(1-P_{1})\right)\right]-a}{b}\right\}$$

The parameters a and b in Equation (13) are obtained from Table 5, and the impact life at different failure probabilities $P_{1}$ can be found by Equation (13), and the estimated impact life of the SPRC at different failure probabilities $P_{1}$ is shown in Table 6.

#### 3.3.2. SPRC Impact-Damage Analysis

In the previous section, the impact life of SPRC with different failure probabilities was predicted using the Weibull probability density function, but the damage variation in the SPRC from the first impact load until impact damage occurred was not systematically studied. In this section, the macroscopic-damage evolution law of each group of specimens after repeated drop-hammer impacts was studied by establishing the SPRC impact-damage model using the Weibull distribution.

With the increase in drop-hammer-impact times, the function value of the Weibull distribution failure probability function also gradually increases, which is an increasing function. From the process of damage deterioration under impact loading, it is clear that the impact-damage degree of the concrete increases with the number of impacts, and with the superposition of impact damage to concrete from each impact load, the fatigue damage from impact loading is a growth process. It can be considered that the damage degree and failure probability accumulate simultaneously in the process of drop-hammer impact.

After n times hammer impacts, the failure probability of the concrete is $P_{F}\left(n\right)$, and the damage degree is $D\left(n\right)$. When the concrete fails by drop-hammer impact, the failure probability of concrete is $P_{F}\left(n\right)=1$, and the damage degree is $D\left(n\right)=1$. In summary, the impact-damage model of concrete based on the Weibull distribution can be expressed by Equation (14):(14)$$D\left(n\right)=1-\exp\left[-{\left(\frac{n}{\eta }\right)}^{\beta }\right]$$
where n is the number of impacts, β and η can be obtained by combining Equations (5)–(10) with Table 7 and substituting the calculation results into Equation (14), and the damage-evolution equation of each group of specimens under the action of impact loading can be derived, as shown in Table 7.

According to the impact-damage-evolution equation in Table 7, the damage evolution of each group of specimens versus impact damage is plotted in Figure 6. The trend of the SPRC impact-damage evolution under repeated impact loadings can be seen. In the early stage of impact damage, the expansion of pores and the extension of microcracks in concrete are small, and the damage deterioration is low. With the increasing number of impacts, the expansion of small pores in concrete and the extension of microcracks increase. The damage and deterioration of the concrete become increasingly severe, and the increase in the loss degree gradually increases. When the loss degree $D\left(N\right)=1$, the concrete is completely destroyed. Figure 6 shows that, in the data results derived from the impact-damage-evolution equation, the impact-damage-life law of concrete is highly consistent with the test results, indicating that the impact-damage-evolution equation established based on the Weibull distribution model can well reflect the impact-damage-evolution trend of SPRC under repeated impact loading.

### 3.4. Analysis of the Impact Life of the SFPC Based on the RSM under Different Failure Probabilities

#### 3.4.1. Factor Level and Response

Response surface method (RSM) is an optimization method that responds to core elements by multiple factors [52]. The functional relationships are displayed graphically for direct observation to select the optimal conditions in an experimental design. The RSM can analyze the obviousness of the test results, obtain the optimal test scheme, and effectively predict the test results. To further explore the effects of silica-fume content, polyvinyl-alcohol-fiber content and failure probability on the impact resistance of SPRC, RSM was used to analyze the impact resistance of SPRC under the interaction of different factors and contents. Considering the single-factor effect and the two-factor interaction effect, the quadratic response surface equation can be established, as shown in Equation (15). The impact life of SPRC was analyzed by RSM with Factor A (silica-fume content), Factor B (polyvinyl-alcohol-fiber content) and Factor C (failure probability). The factors and level variations are shown in Table 8.
(15)$$R=\alpha_{0}+\sum_{i=1}^{n}\alpha_{i}X_{i}+\sum_{i=1}^{n}\alpha_{ii}X_{i}^{2}+\sum_{i<j}^{n}a_{ij}X_{i}X_{j}+\varepsilon \;\;\;(X_{i},X_{i},\;\cdots ,X_{i})$$

In Equation (15), R is the response function (also called the objective function) for independent variable factor $X_{i}$; $\alpha_{i},\;\alpha_{ii},\;\alpha_{ij}$ are the regression coefficients of the primary term coefficient, the secondary term coefficient and the interaction term, respectively; n is the number of influencing factors; and ε is the error (experimental error and fitting error).

#### 3.4.2. Establishment of the RSM Model

RSM analysis design and results are shown in Table 9. The quadratic surface regression model was obtained by fitting multiple regression analysis to the experimental data using Design-Expert software, as shown in Equation (16). Variance analysis was carried out on Equation (16) of the model. The F value and P value of the model are used to test the significance and effectiveness of the model. The F value and P value of the lack of fit are used to test another basis for whether the model is effectiveness. The regression coefficient $R^{2}$ value is used to test the reliability of the model. When the P value of the model is less than 0.05 and the P value of the lack of fit is greater than 0.05, the model has high significance. When the value of the regression coefficient $R^{2}$ is closer to 1 and the C.V. is less than 10%, the model has a high reliability.
(16)$$\begin{array}{ll} R= & 1870.2+24.25A-40.38B+647.88C-4.25AB+\\ & 98.75AC+24BC-186.6A^{2}-395.35B^{2}-301.85C^{2} \end{array}$$

The variance analysis of the impact life response model of the SPRC is shown in Table 10. As seen in Table 10, the P value of the model is much less than 0.05 and the P value of the lack of fit is 0.0696, which is greater than 0.05. This indicates that the model has high significance and effectiveness, reflecting the high correlation between the model and the experimental data. In addition the regression coefficient $R^{2}$ value of the model is 0.9826 and the C.V. value is 7.52%, which indicates that the model has good reliability. Based on the above significance and reliability tests, this model can reasonably describe and predict the impact life of SPRC under the interaction of silica-fume content, polyvinyl-alcohol-fiber content and different failure probabilities.

#### 3.4.3. RSM Model Results Analysis

The RSM scheme was imported into Design-Expert to analyze the effect of the variation in Factor A (silica-fume dosing), Factor B (polyvinyl-alcohol-fiber dosing) and Factor C (failure probability) on the significance of the impact life of SPFC. The contour map and 3D response surface map derived from the model are shown in Figure 7.

Through the analysis of Figure 7a,b, the contour map shows that the surrounding green gradually transforms into orange at the center, and the center is the region with the highest response degree, which indicates that the corresponding material content in this region is the optimal doping value. In the 3D response surface diagram, the trend of the R value along Factor B is greater than that of Factor A, which indicates that Factor B (polyvinyl-alcohol-fiber content) has a higher effect on R values than factor A (silica-fume content).

As seen from Figure 7c,d, in the contour map, it can be seen that the green below gradually changes to the red above. The top right area is the highest response area, and the corresponding failure probability and silica-fume content are the best values. Compared with Factor A, the trend of R value along Factor C is more obvious in the 3D response surface diagram, which indicates that Factor C (failure probability) has greater influence on R values than Factor A (silica-fume content).

As seen from Figure 7e,f, the contour map shows that the green below gradually changes to the red above. The central area above is the area with the highest response, and the corresponding failure probability and polyvinyl-alcohol-fiber content are the best values. Compared with Factor B, the trend of the R value along Factor C is more obvious in the three-dimensional response surface diagram, which indicates that the influence of Factor C (the failure probability) has a greater effect on the R value than that of Factor B (polyvinyl-alcohol-fiber content).

## 4. Conclusions

SPRC was prepared and tested for compressive strength and impact resistance, and the results were analyzed in depth using the Weibull distribution model and RSM. The following conclusions can be drawn from the results obtained:With the increase in silica-fume and polyvinyl-alcohol-fiber content, the compressive strength and impact energy dissipation of SPRC first increase and then decrease. When the silica-fume content is 10% and the polyvinyl-alcohol-fiber content is 1%, the compressive strength and impact resistance of SPRC are the best, and these values are 26.6% and 154.23% higher than those of the benchmark concrete, respectively.The SPRC impact life test results obey the Weibull distribution probability density function well; thus, the Weibull distribution model can be used to effectively predict the impact life of SPRC under different failure probabilities.The equation of the SPRC impact damage evolution is established based on the Weibull distribution model. The predicted values of the model are in high consistency with the experimental results. This indicates that the model has high reliability and can accurately describe the damage degradation process of the SPFC under repeated impact loading.The RSM model was established by combining the level changes of the three factors. The results show that the model has high significance, validity and reliability. Therefore, the impact life of SPRC under the action of these three factors can be effectively predicted.

## Figures and Tables

**Figure 1 polymers-14-02281-f001:**
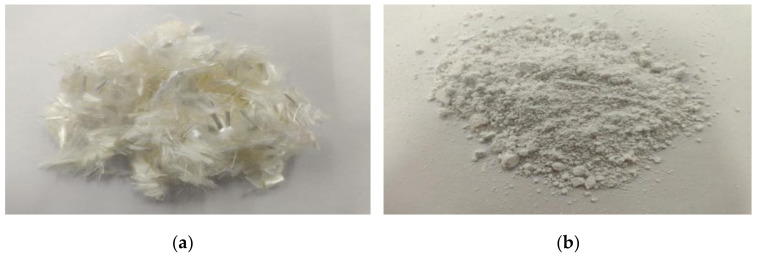
(**a**) Polyvinyl alcohol fiber; (**b**) silica fume.

**Figure 2 polymers-14-02281-f002:**
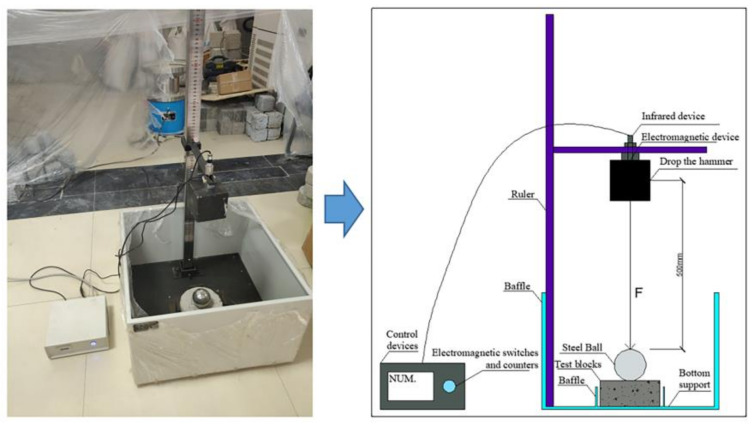
Drop-hammer impact device.

**Figure 3 polymers-14-02281-f003:**
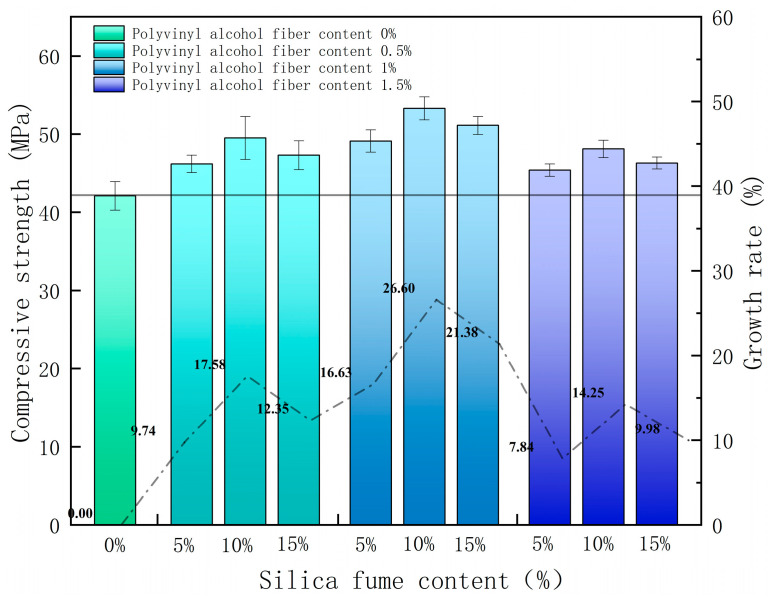
Compressive-strength growth of the SPRC.

**Figure 4 polymers-14-02281-f004:**
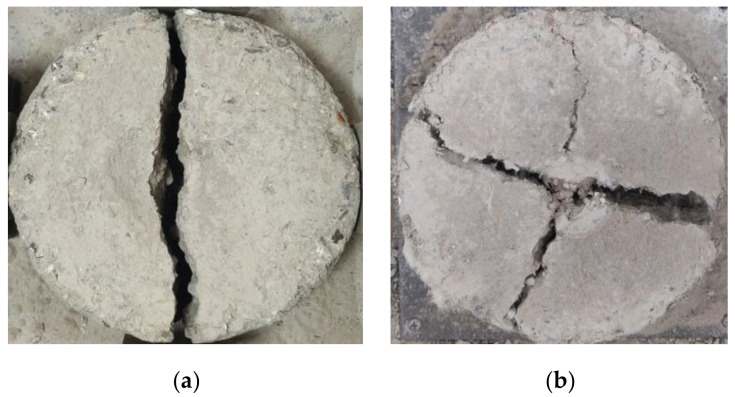
(**a**) Impact rupture form in ordinary concrete; (**b**) impact rupture form in SPRC.

**Figure 5 polymers-14-02281-f005:**
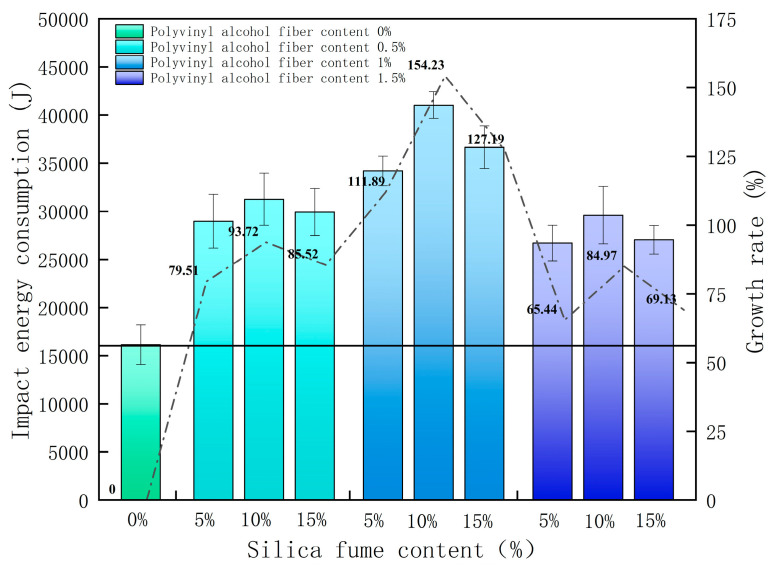
Growth in SPRC impact energy consumption.

**Figure 6 polymers-14-02281-f006:**
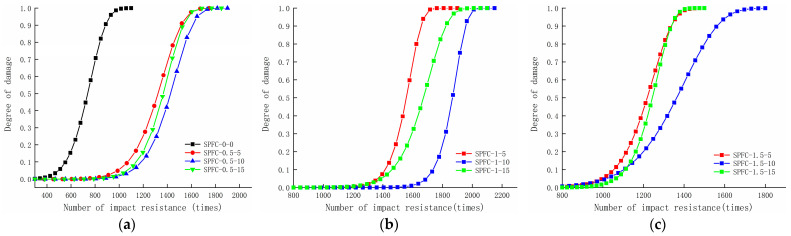
SPRC specimen impact-damage-evolution diagram (**a**) Impact-damage-evolution diagram of specimens with 0 and 0.5% polyvinyl alcohol fiber content; (**b**) Impact-damage-evolution diagram of specimens with 1% polyvinyl alcohol fiber content; (**c**) Impact-damage-evolution diagram of specimens with 1.5% polyvinyl alcohol fiber content.

**Figure 7 polymers-14-02281-f007:**
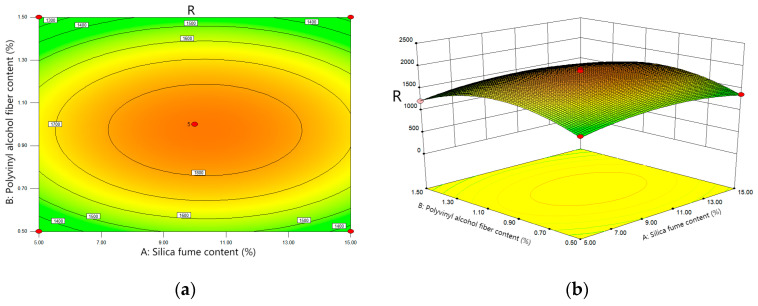
Effects of Factor A, B, C and their interactions on R. (**a**,**b**): Effects of Factor A and B and their interactions on R; (**c**,**d**): Effects of Factor A and C and their interactions on R; (**e**,**f**): Effects of Factor B and C and their interactions on R.

**Table 1 polymers-14-02281-t001:** Detailed parameters of silica fume and polyvinyl alcohol fiber.

Silica fume	SiO_2_ (%)	Loss of ignition (%)	Specific surface area (m^2^/g)	Chloride ion (%)	28 d activity index (%)
	98.1	1.48	21	0.01	105
Fly ash	Fineness (%)	Burning vector (%)	Water content (%)	Density (g/cm^3^)	Bulk density (g/cm^3^)
	16	2.8	0.85	2.55	1.12
Cement	Standard consistency water consumption (%)	Initial setting time (min)	Final setting time (min)	Cube compressive strength (MPa)	
				7 d	28 d
	26.6	196	265	25.8	44.5
Polyvinyl alcohol fiber	Fiber diameter (μm)	Fiber length (mm)	Tensile strength (MPa)	Elongation at break (%)	Cross-sectional expansion rate (cN/dtex)
	31	9	1400–1600	17 ± 3.0	320

**Table 2 polymers-14-02281-t002:** Mix proportions of the designed concrete.

NO.	Polyvinyl Alcohol Fiber/%	Content of Material (kg/m^3^)						
		Silica Fume	Cement	Coarse Aggregate	Sand	Fly Ash	Water	Water-Reducing Agent
SPRC-0-0 *	-	-	330	1061	837	61	170	1
SPRC-0.5-5	0.5	16.5	313.5	1061	837	61	170	1
SPRC-0.5-10		33	297	1061	837	61	170	1
SPRC-0.5-15		49.5	280.5	1061	837	61	170	1
SPRC-1-5	1	16.5	313.5	1061	837	61	170	1
SPRC-1-10		33	297	1061	837	61	170	1
SPRC-1-15		49.5	280.5	1061	837	61	170	1
SPRC-1.5-5	1.5	16.5	313.5	1061	837	61	170	1
SPRC-1.5-10		33	297	1061	837	61	170	1
SPRC-1.5-15		49.5	280.5	1061	837	61	170	1

* SPRC-0-0 means that the content of silica fume is 0 and the volume content of polyvinyl alcohol fiber is 0. The latter groups, and so on.

**Table 3 polymers-14-02281-t003:** $N_{1}$ and $N_{2}$ under concrete drop-hammer impact.

NO.	$N_{1}/N_{2}$					
	1	2	3	4	5	6
SPRC-0-0	574/575	733/734	674/676	788/789	812/814	805/806
SPRC-0.5-5	1278/1288	1156/1168	1380/1388	1403/1410	1448/1462	1153/1166
SPRC-0.5-10	1356/1372	1212/1226	1577/1591	1425/1436	1384/1393	1478/1491
SPRC-0.5-15	1231/1242	1422/1435	1187/1201	1376/1384	1403/1412	1457/1472
SPRC-1-5	1643/1657	1571/1582	1432/1449	1527/1542	1554/1567	1495/1510
SPRC-1-10	1908/1928	1856/1873	1744/1762	1872/1887	1793/1810	1891/1907
SPRC-1-15	1778/1803	1603/1616	1677/1687	1568/1580	1732/1748	1530/1542
SPRC-1.5-5	1248/1258	1321/1330	1156/1168	1227/1238	1176/1189	1078/1085
SPRC-1.5-10	1474/1488	1531/1543	1243/1256	1278/1289	1341/1353	1185/1193
SPRC-1.5-15	1315/1327	1144/1154	1287/1296	1173/1185	1248/1257	1195/1207

**Table 4 polymers-14-02281-t004:** Impact resistance index analysis results.

NO.	Average Number of Impact Resistance			*λ* (%)	*W* (J)	Growth Rate of *W* (%)
	$N_{1}$	$N_{2}$	$N_{2}-N_{1}$			
SPRC-0-0	731	732	1	0.14	16,157.07	0.00
SPRC-0.5-5	1303	1314	11	0.84	29,003.27	79.51
SPRC-0.5-10	1405	1418	13	0.93	31,298.81	93.72
SPRC-0.5-15	1346	1358	12	0.89	29,974.46	85.52
SPRC-1-5	1537	1551	14	0.91	34,234.45	111.89
SPRC-1-10	1844	1861	17	0.92	41,076.92	154.23
SPRC-1-15	1648	1663	15	0.91	36,706.57	127.19
SPRC-1.5-5	1201	1211	10	0.83	26,729.80	65.44
SPRC-1.5-10	1342	1354	12	0.89	29,886.17	84.97
SPRC-1.5-15	1227	1238	11	0.90	27,325.76	69.13

**Table 5 polymers-14-02281-t005:** Regression parameter values.

NO.	Regression Parameter		Regression Coefficient
SPRC-0-0	a=−43.95136	b=6.59925	$R^{2}=0.91731$
SPRC-0.5-5	a=−64.91919	b=8.98057	$R^{2}=0.89184$
SPRC-0.5-10	a=−75.09988	b=10.28831	$R^{2}=0.95511$
SPRC-0.5-15	a=−77.32255	b=10.65826	$R^{2}=0.9082$
SPRC-1-5	a=−147.07487	b=19.95748	$R^{2}=0.95815$
SPRC-1-10	a=−199.48323	b=26.43486	$R^{2}=0.95857$
SPRC-1-15	a=−104.51039	b=14.03187	$R^{2}=0.82022$
SPRC-1.5-5	a=−92.86202	b=13.01777	$R^{2}=0.97532$
SPRC-1.5-10	a=−64.37724	b=8.86828	$R^{2}=0.91335$
SPRC-1.5-15	a=−119.01573	b=16.65037	$R^{2}=0.951$

**Table 6 polymers-14-02281-t006:** Impact life of each group of specimens under different failure probabilities (times).

NO.	$P_{1}=0.3$	$P_{1}=0.4$	$P_{1}=0.5$	$P_{1}=0.6$	$P_{1}=0.7$
SPRC-0-0	668	705	738	770	803
SPRC-0.5-5	1229	1279	1324	1365	1407
SPRC-0.5-10	1339	1386	1428	1467	1507
SPRC-0.5-15	1284	1328	1367	1403	1440
SPRC-1-5	1507	1534	1558	1580	1602
SPRC-1-10	1821	1846	1867	1887	1907
SPRC-1-15	1595	1636	1672	1706	1739
SPRC-1.5-5	1158	1190	1218	1245	1271
SPRC-1.5-10	1265	1318	1364	1407	1451
SPRC-1.5-15	1195	1221	1244	1265	1286

**Table 7 polymers-14-02281-t007:** SPRC impact-damage-evolution equation.

$SPFC-0-0:D\left(n\right)=1-\exp\left[-{\left(\frac{n}{780.593}\right)}^{6.599}\right]$	$SPFC-0.5-5:D\left(n\right)=1-\exp\left[-{\left(\frac{n}{1378.636}\right)}^{8.981}\right]$
$SPFC-0.5-10:D\left(n\right)=1-\exp\left[-{\left(\frac{n}{1479.612}\right)}^{10.288}\right]$	$SPFC-0.5-15:D\left(n\right)=1-\exp\left[-{\left(\frac{n}{1414.748}\right)}^{10.658}\right]$
$SPFC-1-5:D\left(n\right)=1-\exp\left[-{\left(\frac{n}{1586.699}\right)}^{19.957}\right]$	$SPFC-1-10:D\left(n\right)=1-\exp\left[-{\left(\frac{n}{1893.569}\right)}^{26.435}\right]$
$SPFC-1-15:D\left(n\right)=1-\exp\left[-{\left(\frac{n}{1716.552}\right)}^{14.032}\right]$	$SPFC-1.5-5:D\left(n\right)=1-\exp\left[-{\left(\frac{n}{1253.232}\right)}^{13.018}\right]$
$SPFC-1.5-10:D\left(n\right)=1-\exp\left[-{\left(\frac{n}{1421.219}\right)}^{8.87}\right]$	$SPFC-1.5-15:D\left(n\right)=1-\exp\left[-{\left(\frac{n}{1271.475}\right)}^{16.65}\right]$

**Table 8 polymers-14-02281-t008:** Factor and factor level.

Factor	Factor Level		
	−1	0	1
Silica fume content (%)	5	10	15
Polyvinyl alcohol fiber content (%)	0.5	1	1.5
Probability of failure	0	0.5	1

**Table 9 polymers-14-02281-t009:** RSM analysis design and results.

NO.	Experimental Design			Results/Times
	A/%	B/%	C	
1	5.00	1.00	0.00	891
2	10.00	1.00	0.50	1867
3	5.00	0.50	0.50	1324
4	10.00	1.00	0.50	1928
5	5.00	1.00	1.00	1810
6	5.00	1.50	0.50	1218
7	15.00	1.00	0.00	756
8	15.00	1.00	1.00	2070
9	10.00	1.00	0.50	1887
10	10.00	1.00	0.50	1762
11	15.00	1.50	0.50	1244
12	10.00	0.50	1.00	1910
13	10.00	1.50	1.00	1911
14	10.00	1.00	0.50	1907
15	15.00	0.50	0.50	1367
16	10.00	1.50	0.00	388
17	10.00	0.50	0.00	483

**Table 10 polymers-14-02281-t010:** Variance analysis of the SPRC impact-life response model.

Source	Sun of Squares	Mean of Squares	F Value	*p* Value
Model	4.73 × 10^6^	5.225 × 10^5^	43.99	<0.0001
A	4704.5	4704.5	0.39	0.5502
B	13,041.13	13,041.13	1.09	0.3308
C	3.358 × 10^6^	3.358 × 10^6^	281.08	<0.0001
AB	72.25	72.25	6.048 × 10^−3^	0.9402
AC	39,006.25	39,006.25	3.27	0.1137
BC	2304	2304	0.19	0.6738
A2	1.466 × 10^5^	1.466 × 10^5^	12.27	0.0100
B2	6.581 × 10^5^	6.581 × 10^5^	55.09	0.0001
C2	3.836 × 10^5^	3.836 × 10^5^	32.11	0.0008
Lack of Fit	66,931.25	22,310.42	5.35	0.0696

## Data Availability

The data used to support the findings of this study are available from the corresponding author upon request.

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
