# Peer review of "Study of SPRC Impact Resistance Based on the Weibull Distribution and the Response Surface Method"

_polymers, 2022, doi:10.3390/polym14112281_

Round 1
Reviewer 1 Report
The article presents an analysis of compressive strength and impact resistance tests of green concrete containing microsilica and polyvinyl alcohol fibers. An extensive statistical analysis of the results was performed, using various computational models. In order to improve the quality of the article, the following comments should be taken into account:
-Line 48: In my opinion, it should be “pozzolanic activity”, not “volcanic ash activity”.
-Line 63-64: "with simultaneous volcanic ash" - not microsilica but volcanic ash?
-In the last paragraph of the introduction, the scope of the article should be defined more precisely - to mention, for example, the methodology contained in the title of the manuscript.
- provide details of the origin of the raw materials used.
- why the fly ash parameters were not given in the same way as the silica fume parameters? Cement parameters can also be included in the same way
- a photo of the fibers used would be useful
-line 113: is it the proportion by weight in relation to the cement?
- line 115: are the proportions by volume in relation to the volume of the entire concrete mix or cement?
- Table 2: not stone, but gravel
- the same w/b and the same amount of water were used. Was there no change in the consistency of the concrete mix with the use of larger amounts of microsilica? It is more water-hungry than cement. Were there problems with compaction?
- line 129-131 - a bit strange writing style. It sounds like an instruction, a command. When describing the procedures in the articles, it is rather written that the authors performed these activities, that is, in the past simple, perfect. In general, there is such a style throughout the article. I propose to convert to the past form.
- line 152 - "... ball ball ..." - duplication of a word
-Figure 1 - the font in the diagrams is very small. On the other hand, the diagram of the press seems to be redundant. This is standard in concrete research, and I think the interested reader knows the principle of test, and besides, the same is in the photo on the left.
- There are error bars in the bar charts. Please write if they are. Is this a standard deviation?
- Figure 2 - do “growth rate” values ​​have to be accurate to 4 decimal places? In Table 4, the growth rates are accurate to 2 decimal places.
- Figure 3 - wrong graph has been inserted
- There are no photos showing images of the sample damages after tests.
Reviewer 2 Report
High-performance and ultra-high-performance fiber-reinforced concrete (UHPFRC) can be used to retrofit existing structures to improve their properties. One such application is the use of fiber-reinforced concrete to increase the impact resistance of some structures.
The proposed manuscript exposes a ‘Study of SPRC impact resistance based on Weibull distribution
and response surface method’. The organization of the manuscript is clear and relatively easy to read, though some English proofreading might be helpful. This kind of article might be of interest to the readers of the journal, but the novelty of the research work must be clearly exposed. Many studies dealing with the impact resistance of UHPFRC have been published (most of them including silica fume), but this kind of study is not reported in the introduction, which then alters the perceived novelty of the study.
Aside from this important comment, several other comments need to be addressed:
l 27-28: the sentence is missing a verb
l 39: ‘have added aggregates’: agg are normally present in concrete. Please precise how changing the aggregate content can change the impact resistance
l 41-42: please give more details about the results from the literature
l 46-53: the paragraph about silica fume effect should be revised to include typical cement replacement ratios by SF, and some drawbacks of silica fume: one of the most critical being the increase of shrinkage (autogenous, chemical) due to silica fume, especially when being used in Ultra high performance concrete (UHPC), for example
l 91-96: studies about concrete incorporating silica fume and PVA fibers must be reviewed (especially high performance concrete and UHPC). This type of study is relatively common. Therefore, the novelty of the manuscript is not sufficiently argued.
Materials and methods
l 108: precise meaning of C40 concrete (as it might differ from one standard to another, especially concerning the shape of the specimens)
l 109 – 112: revise infinitive sentence and remove ‘Take’
Table 2: please use ‘Coarse aggregate’ rather than ‘Stone’
Table 2: please precise if the water reducing agent is 1% cement mass or 1 kg/m3
l 122: please explain how the fibers were dispersed
l 141: please remove unnecessary details such as ‘start the machine’, etc
Conclusion
please introduce the conclusions
Reviewer 3 Report
Polymers 1739262
The material silica fume-polyvinyl alcohol (SFRC) is a green and environmentally friendly composite material. Study the impact resistance of material, compressive strength and drop hammer impact tests were conducted on SPRC with different silica fume and polyvinyl alcohol contents.
In this paper, the authors were analyzed mechanical and impact resistance. The impact life of SFRC under the action of silica fume content, polyvinyl fiber content was analyzed in depth by RSM response model. The results indicate that content of silica fume is 10% and polyvinyl alcohol fiber is 1%.
This paper is of sufficiently quality for publication in Polymers. The experimental, results analysis, study and information that the authors show is very interesting but is necessary that the authors check the writing of the manuscript.
Comment
1) The part 3.4, so much the information as the writing is very correct and help to understand all manuscript. Rewrite the manuscript considerer the presentation and writing this part.
2) In this manuscript, the authors repeat mean times the word “concrete”. If possible, delete ever of the words.
Other Comment
1) Define the letters RSM. Revise all manuscript
2) Line 47-90, rewrite the paragraph. It is confusing and hard to read
3) Line 95, delete “effect”
4) Line 100, change 25mm for 25 mm. Revise all manuscript
5) Table 1, change chlorine ion for chloride ion
6) Line 113, delete “were added to the concrete”
7) Line 146, change [47], for [47]. Revise all manuscript
8) Line 151, change Ф63 mm for Ф 63 mm
9) Line 170, delete “particles”
10) Line 173, delete “of concrete”
11) Line 177, change region.Silice for region. Silice
12) Line 189, delete “the concrete”
13) Revise the part 3.2 Impact resistance of SPRC
14) Line 265, delete “model”
15) Β: Is used as: Ductility ration and, shape factor. The use the same letter for two different parameters can lead to confusion
16) Line 357, change D(n1).when for D(n1). When. Revise all manuscript
17) Line 356-360, define n1 and N. As written is confusing
18) Line 472, change decrease When for decrease. When. Revise all manuscript
19) Revise the conclusions 3 and 4
20) In abstract and conclusions used “RSM” but these letters are not defined in the manuscript
Round 2
Reviewer 2 Report
The authors revised their manuscript based on the various comments from the reviewers. The introduction and the literature background have been amended. More discussions are provided.
In my opinion, the additional elements provided by the authors are sufficient regarding journal’s scope and, hopefully, the article may be of interest to the readers of the journal.
Reviewer 3 Report
With the changes introduced by the authors in the manuscript. The paper can be accepted in Polymers, in present form